# Disruption of the grid cell network in a mouse model of early Alzheimer's disease

Johnson Ying[1,2], Alexandra T. Keinath [1], Raphael Lavoie[1], Erika Vigneault[1], Salah El Mestikawy [1,2] & Mark P. Brandon [1,2 ✉]

Early-onset familial Alzheimer's disease (AD) is marked by an aggressive buildup of amyloid beta (Aβ) proteins, yet the neural circuit operations impacted during the initial stages of Aβ pathogenesis remain elusive. Here, we report a coding impairment of the medial entorhinal cortex (MEC) grid cell network in the J20 transgenic mouse model of familial AD that over-expresses Aβ throughout the hippocampus and entorhinal cortex. Grid cells showed reduced spatial periodicity, spatial stability, and synchrony with interneurons and head-direction cells. In contrast, the spatial coding of non-grid cells within the MEC, and place cells within the hippocampus, remained intact. Grid cell deficits emerged at the earliest incidence of Aβ fibril deposition and coincided with impaired spatial memory performance in a path integration task. These results demonstrate that widespread Aβ-mediated damage to the entorhinal-hippocampal circuit results in an early impairment of the entorhinal grid cell network.

[1] Department of Psychiatry, Douglas Hospital Research Centre, McGill University, Montreal, QC, Canada. [2] Integrated Program in Neuroscience, McGill University, Montreal, QC, Canada. ✉email: mark.brandon@mcgill.ca

The molecular and synaptic underpinnings of Aβ pathology during the earliest stages of familial AD are well-documented, but the impact that these changes have on neural coding has not been resolved[1,2]. The emergence of spatial memory deficits in patients with preclinical AD and those with Mild Cognitive Impairment with high levels of cerebrospinal fluid Aβ suggest that Aβ pathology exerts its earliest impact on the neural systems that support spatial memory[3,4]. Extensive work in both animals and humans pinpoint the MEC-hippocampal circuit as an essential brain region for spatial memory performance[5–7]. At the level of neural coding, the MEC-hippocampal circuit contains a myriad of spatially tuned cell types including place cells in the hippocampus, as well as grid cells, head-direction cells, and non-grid spatially selective cells in the MEC[8–12]. Decades of theoretical work have proposed how these functional cell types work in concert to support spatial memory[13–17]. Yet, it remains unknown how these spatially tuned populations are impacted during the earliest stages of Aβ-mediated pathogenesis when spatial memory is impaired.

To address this question, we recorded spatially tuned neurons from the hippocampus and MEC of the J20 transgenic mouse model of familial AD that expresses a mutant form of the human amyloid precursor protein (APP), referred to here as 'APP mice'[18]. In this model, elevated and comparable levels of soluble Aβ throughout the entorhinal cortex and hippocampus are present at 3 months of age[19]. By 5–7 months of age, small Aβ fibrils are detectable in the hippocampus but neither of these regions demonstrate widespread amounts of Aβ plaques that are indicative of late AD pathology[19]. We confirmed that APP mice expressed little-to-no plaques by 6 months of age in the MEC and hippocampus (Supplementary Fig. 1). Between 3 and 7 months of age, APP mice exhibit several amyloid-related processes that we refer to collectively as 'early Aβ pathology'. These include neuroinflammation, 10–20% neuronal loss, and reduced presynaptic terminal density throughout the entorhinal cortex and hippocampus (detailed pathology description in Methods, Subjects)[20–23].

In this work, we show that early Aβ pathology reduces grid cell spatial coding in an age-dependent manner preceding the widespread expression of Aβ plaques. In contrast, the spatial coding of non-grid cells within the MEC, and place cells within the hippocampus, is unaffected. The grid cell impairments correlate with worsened spatial navigation performance in a path integration task, thus pointing to both grid cell integrity and path integration performance as possible early markers of AD in familial and sporadic populations.

## Results

**Grid cell spatial tuning in APP mice is impaired across age.** We obtained in vivo recordings of MEC neurons ($n$ cells = 4524) from 38 APP transgenic and 30 non-transgenic (nTG) littermates as they foraged for water droplets in an open field arena (Summary of MEC recordings, Supplementary Table 1; MEC Tetrode locations, Supplementary Fig. 2). We observed an age-related disruption in the spatial periodicity of grid cells in APP mice (Fig. 1a–d, Supplementary Figs. 3, 4, 5). Young APP mice (3–4.5 months) had grid cells with tuning comparable to those of age-matched nTG mice (Fig. 1). In contrast, grid cells recorded in adult (4.5–7 months) APP mice exhibited reduced spatial periodicity and spatial information (bits/spike) in comparison to those from young APP mice and age-matched nTG mice (Fig. 1). Peak spatial firing and mean firing rates of grid cells did not reliably differ between groups and across age (Fig. 1e). A two-way ANOVA was conducted to determine the effects of age and genotype on grid scores between groups. A significant interaction effect was discovered, supporting the view that grid cell spatial

periodicity is reduced across age in APP mice (ANOVA, age main effect: $P = 1.89 \times 10^{-7}$; genotype main effect: $P = 0.011$; interaction effect: $P = 3.86 \times 10^{-4}$, Supplementary Fig. 6). To ensure that these results are not biased by oversampling the same cells across days, we removed duplicate grid cells and re-ran our analyses. Grid cell spatial periodicity remained impaired in adult APP mice, and the significant interaction effect persisted (ANOVA, age main effect: $P = 4.5 \times 10^{-4}$; genotype main effect: $P = 0.013$; interaction effect: $P = 0.046$, Supplementary Fig. 7).

**Spatial tuning of non-grid MEC cells remains intact in APP mice.** In contrast to the age-dependent impairment observed in grid cells, entorhinal head-direction cells, which encode the orientation of the animal's head in polar coordinates[10], did not differ in their directional tuning or firing rates between groups or across age (Fig. 2a, Supplementary Fig. 8). Similarly, there was no difference between groups in the average firing field size of non-grid spatially tuned neurons, which fire in a non-periodic but spatially reliable manner (Fig. 2b, Supplementary Fig. 8). Mean firing rates did not differ between groups, but spatial peak firing rates were, however, oddly elevated in adult nTG mice (Fig. 2b).

To examine if spatial coding by downstream hippocampal place cells was disrupted when adult APP mice exhibit a degraded grid cell code, we obtained in vivo recordings from region CA1 of the hippocampus ($n$ cells = 992) from 6 adult APP and 6 adult nTG mice (Summary of CA1 recordings, Supplementary Table 2; CA1 Tetrode locations, Supplementary Fig. 9). Place cells in adult APP and nTG mice were similarly tuned for spatial location and had similar peak spatial and mean firing rates (Fig. 2c, Supplementary Fig. 8). Spatial tuning remained largely preserved across groups when varying our cell selection threshold (peak spatial firing rates between 0 and 8 Hz), with the exception of thresholds less than 1 Hz (Fig. 2d). Mean firing rates were higher in APP place cells at peak firing selection thresholds of 6 Hz and greater, suggesting that the overall mean firing rate is higher in adult APP mice than those in adult nTG mice (Fig. 2d). These results demonstrate that the hippocampal place code remains grossly intact when the entorhinal grid code is degraded in adult APP mice, mirroring findings observed in early development and during inactivation of the medial septum[24–26]. Our findings are consistent with previous work showing that the spatial tuning of place cells in Tg2576 APP mice remained intact during the earliest incidence of Aβ plaques, but was subsequently impaired when Aβ plaques become widespread[27]. Moreover, our results suggest that impaired grid coding in adult APP mice is not the result of disrupted feedback from the hippocampus[28]. Prior work has reported a selective disruption of grid cell spatial periodicity without impairment of other spatial codes when the power of entorhinal theta oscillations (6–10 Hz) is reduced via inactivation of the medial septum[25,26]. We therefore examined entorhinal theta oscillations across nTG and APP mice and found that theta power remained intact in adult APP mice across running speeds (ANCOVA, APP-a vs nTG-a, main effect: $P = 0.99$; interaction between running speed and theta power: $P = 0.096$; APP-a vs APP-y, main effect: $P = 0.051$; interaction between running speed and theta power: $P = 0.56$; Fig. 2e). In both young and adult APP mice, the overall baseline frequency of theta oscillations was lower while the gain in theta frequency across running speeds was preserved (ANCOVA, APP-a vs nTG-a, main effect: $P = 0.01$; interaction between running speed and frequency: $P = 0.51$; APP-y vs nTG-y, main effect: $P = 0.0001$; interaction between running speed and frequency: $P = 0.82$; Fig. 2e), revealing that the baseline frequency of theta oscillations was reduced in APP mice prior to the onset of grid cell disruption. Theta frequency reduction in both young and adult APP mice was roughly 0.2 Hz across all

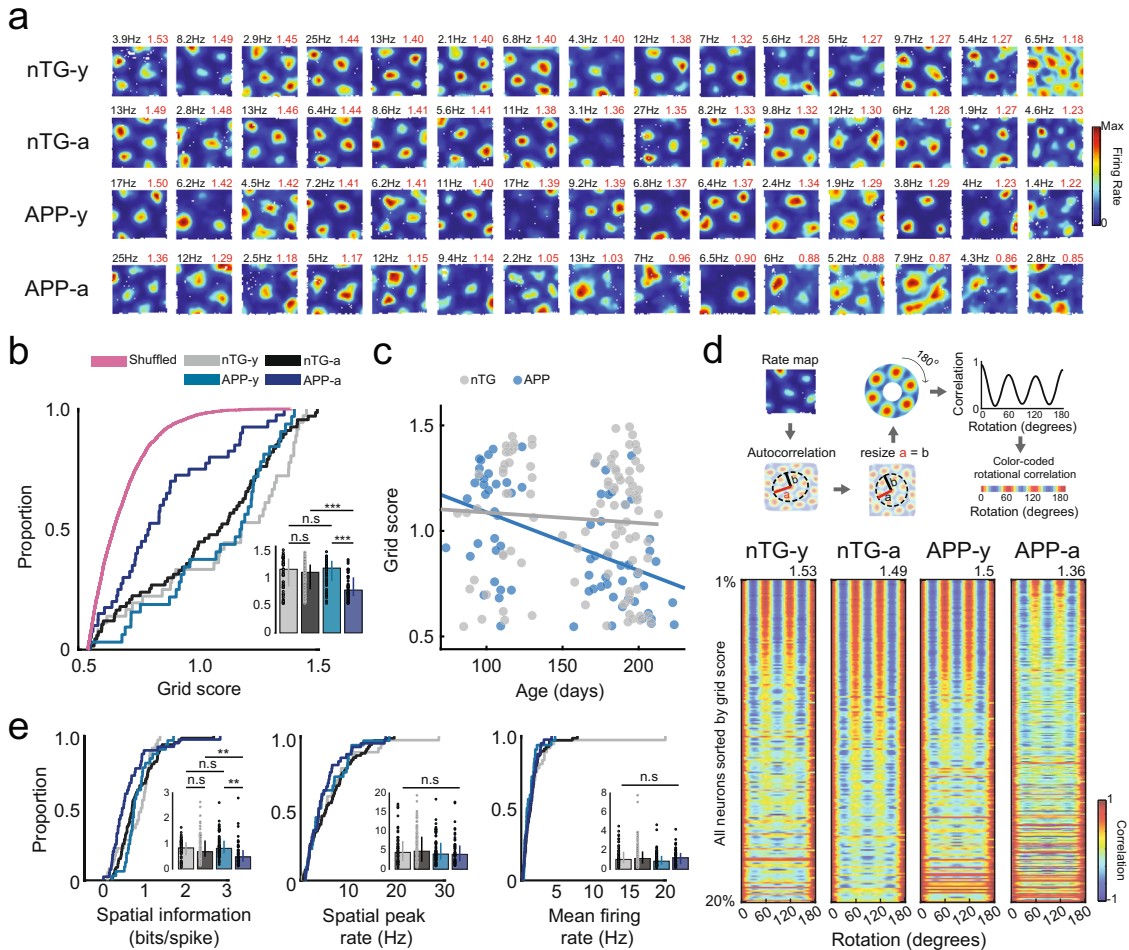

**Fig. 1 Reduction of grid cell spatial periodicity in adult APP transgenic mice. a** Firing rate maps for grid cells from each experimental group. Each row includes 15 grid cells with the highest grid scores sorted in descending order. Spatial peak firing rate and grid score are indicated in the rate map's top-left and top-right, respectively. **b** Grid scores (nTG-y-nTG-a: $P = 0.12$; nTG-y-APP-y: $P = 0.73$; APP-y-APP-a: $P = 1 \times 10^{-7}$; nTG-a-APP-a: $P = 2.5 \times 10^{-6}$) between groups (cells, $n = 64$ nTG-y; $n = 99$ nTG-a; $n = 74$ APP-y; $n = 50$ APP-a). **c** Scatter plot displays grid score by age (in days) recorded. A two-way ANOVA examined the effects of age and genotype on grid score. There was a significant interaction between age and genotype: $F(1, 280) = 11.99$, $P = 6.2 \times 10^{-4}$. **d** Color-coded rotational correlations are shown, sorted in descending order of grid score. Neurons within the top 20% of grid scores are shown. The max grid score in each group is displayed at the top of the respective plot. **e** Spatial information (nTG-y-nTG-a: $P = 0.77$; nTG-y-APP-y: $P = 0.34$; APP-y-APP-a: $P = 6.5 \times 10^{-6}$; nTG-a-APP-a: $P = 1.2 \times 10^{-3}$), spatial peak firing rate (nTG-y-nTG-a: $P = 0.59$; nTG-y-APP-y: $P = 0.79$; APP-y-APP-a: $P = 0.77$; nTG-a-APP-a: $P = 0.27$), and mean firing rate (nTG-y-nTG-a: $P = 0.99$; nTG-y-APP-y: $P = 0.086$; APP-y-APP-a: $P = 0.13$; nTG-a-APP-a: $P = 0.79$) between groups (cells, $n = 64$ nTG-y; $n = 99$ nTG-a; $n = 74$ APP-y; $n = 50$ APP-a). nTG-y non-transgenic young, nTG-a non-transgenic adult, APP-y APP young, APP-a APP adult. Wilcoxon rank-sum tests (two-sided) corrected for multiple comparisons using a Bonferroni-Holm correction were applied to analyze the data in panels **b** and **e**. Data in bar graphs are presented as median values ± 25th and 75th percentiles; **$P < 0.01$, ***$P < 0.001$; n.s not significant. Source data are provided as a Source Data file.

running speeds (Supplementary Fig. 10). Assuming that this reduction in theta frequency has no effect on grid cell periodicity in young APP mice, these results indicate that impaired grid cell coding in adult APP mice cannot be explained by a disruption of the theta-generating circuit.

Speed cells in the MEC encode the animal's running speed by firing rate and are assumed to provide a speed signal for grid cell formation[29]. To determine if impaired speed cells could explain the disrupted grid cell periodicity in adult APP mice, we examined the running speed vs firing rate correlation of MEC cells that were not characterized as either grid cells, head-direction cells and non-grid spatially tuned cells (Supplementary Fig. 11). No significant differences were found between groups when varying our cell selection threshold (running speed vs firing rate correlation values between 0.1 and 0.9), suggesting that MEC speed cells remained unaffected by APP pathology (Supplementary Fig. 11a). Running

speed vs firing rate correlations of grid cells were also non-significant between groups, providing further evidence for an intact speed code (Supplementary Fig. 11b, c).

**Grid cells in adult APP mice are spatially unstable**. To characterize the nature of reduced grid cell periodicity in adult APP mice, we examined the spatial firing properties of grid cells in further detail. In contrast to grid cells recorded in adult nTG mice, grid cells in adult APP mice exhibited larger firing fields when accounting for differences in spatial scale (Fig. 3a). We reasoned that an increase in field size in adult APP mice could reflect a drifting or unstable grid pattern over time. Consistent with this idea, a partitioned rate map stability analysis revealed that the overall grid pattern in adult APP mice exhibited reduced spatial stability (Fig. 3b–d). The reduced stability of grid cells in

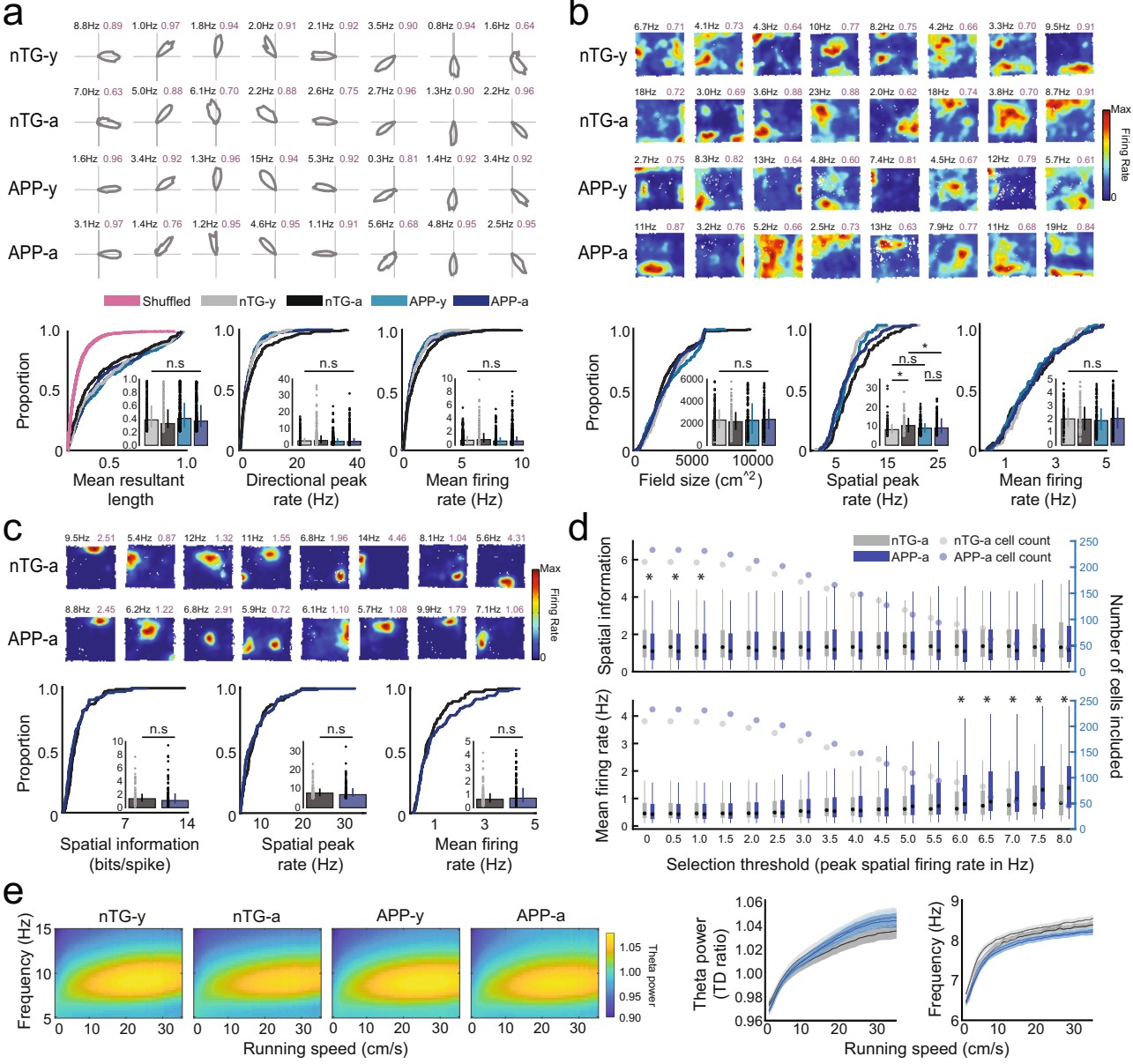

**Fig. 2 Head-direction cells, non-grid spatially tuned cells, place cells, and MEC theta oscillations in APP mice. a** Head-direction polar plots; peak firing rate (F.R) (top-left), mean resultant length (MRL) (top-right). MRL (nTG-y-nTG-a: $P = 0.051$; nTG-y-APP-y: $P = 0.45$; APP-y-APP-a: $P = 0.41$; nTG-a-APP-a: $P = 0.11$), peak F.R (nTG-y-nTG-a: $P = 0.44$; nTG-y-APP-y: $P = 0.21$; APP-y-APP-a: $P = 0.96$; nTG-a-APP-a: $P = 0.073$), mean F.R (nTG-y-nTG-a: $P = 0.18$; nTG-y-APP-y: $P = 0.28$; APP-y-APP-a: $P = 0.87$; nTG-a-APP-a: $P = 0.08$). $n = 295$ nTG-y; 244 nTG-a; 305 APP-y; 471 APP-a. **b** Non-grid spatially tuned rate maps; peak F.R (top-left), split-half-reliability (top-right). Field size (nTG-y-nTG-a: $P = 0.40$; nTG-y-APP-y: $P = 0.66$; APP-y-APP-a: $P = 0.64$; nTG-a-APP-a: $P = 0.37$), peak F.R (nTG-y-nTG-a: $P = 0.028$; nTG-y-APP-y: $P = 0.49$; APP-y-APP-a: $P = 0.73$; nTG-a-APP-a: $P = 0.032$), mean F.R (nTG-y-nTG-a: $P = 0.76$; nTG-y-APP-y: $P = 0.37$; APP-y-APP-a: $P = 0.50$; nTG-a-APP-a: $P = 0.93$). $n = 82$ nTG-y; 83 nTG-a; 106 APP-y; 119 APP-a. **c** Place cell rate maps; peak F.R (top-left), spatial information (S.I) (top-right). S.I ($P = 0.08$), peak F.R ($P = 0.32$), mean F.R ($P = 0.31$). $n = 118$ nTG-a; 109 APP-a. **d** S.I ($P$ left-to-right: 0.03; 0.03; 0.028; 0.057; 0.10; 0.15; 0.16; 0.28; 0.18; 0.12; 0.08; 0.16; 0.11; 0.12; 0.30; 0.31; 0.41), mean F.R ($P$ left-to-right: 0.64; 0.64; 0.70; 0.81; 0.95; 0.99; 0.90; 0.82; 0.85; 0.73; 0.31; 0.24; 0.048; 0.016; 0.025; 0.027; 0.039) when varying peak F.R criterion. (nTG-a $n$ left-to-right: 210; 210; 209; 205; 197; 187; 172; 159; 147; 136; 118; 103; 91; 83; 75; 61; 56. APP-a $n$ left-to-right: 233; 233; 231; 224; 212; 202; 185; 166; 148; 127; 109; 94; 77; 62; 54; 49; 45) **e** MEC theta frequency and power versus running speed. Data are mean ± 99% confidence intervals. Wilcoxon rank-sum tests (two-sided) with Bonferroni-Holm's correction were applied to panels **a–d**. Data in bar graphs are medians ± 25th and 75th percentiles; Boxplots present medians as dots, interquartile boxes from 25th to 75th percentile, whiskers from smallest to largest values; *$P < 0.05$; n.s not significant. Source data are provided as a Source Data file.

adult APP mice was not due to changes in the orientation of grid fields, indicating that instability reflects an inconsistent spatial phase of the overall grid pattern over time (Fig. 3e). In contrast, non-grid spatially tuned cells and hippocampal place cells of adult APP mice remained spatially stable across time (Fig. 3b–d). A

two-way ANOVA was conducted to further confirm that the spatial instability was specific to grid cells, but not non-grid spatially tuned cells and place cells in adult APP mice. The ANOVA design's factors consisted of genotype and cell type, and both significant main and interaction effects were discovered

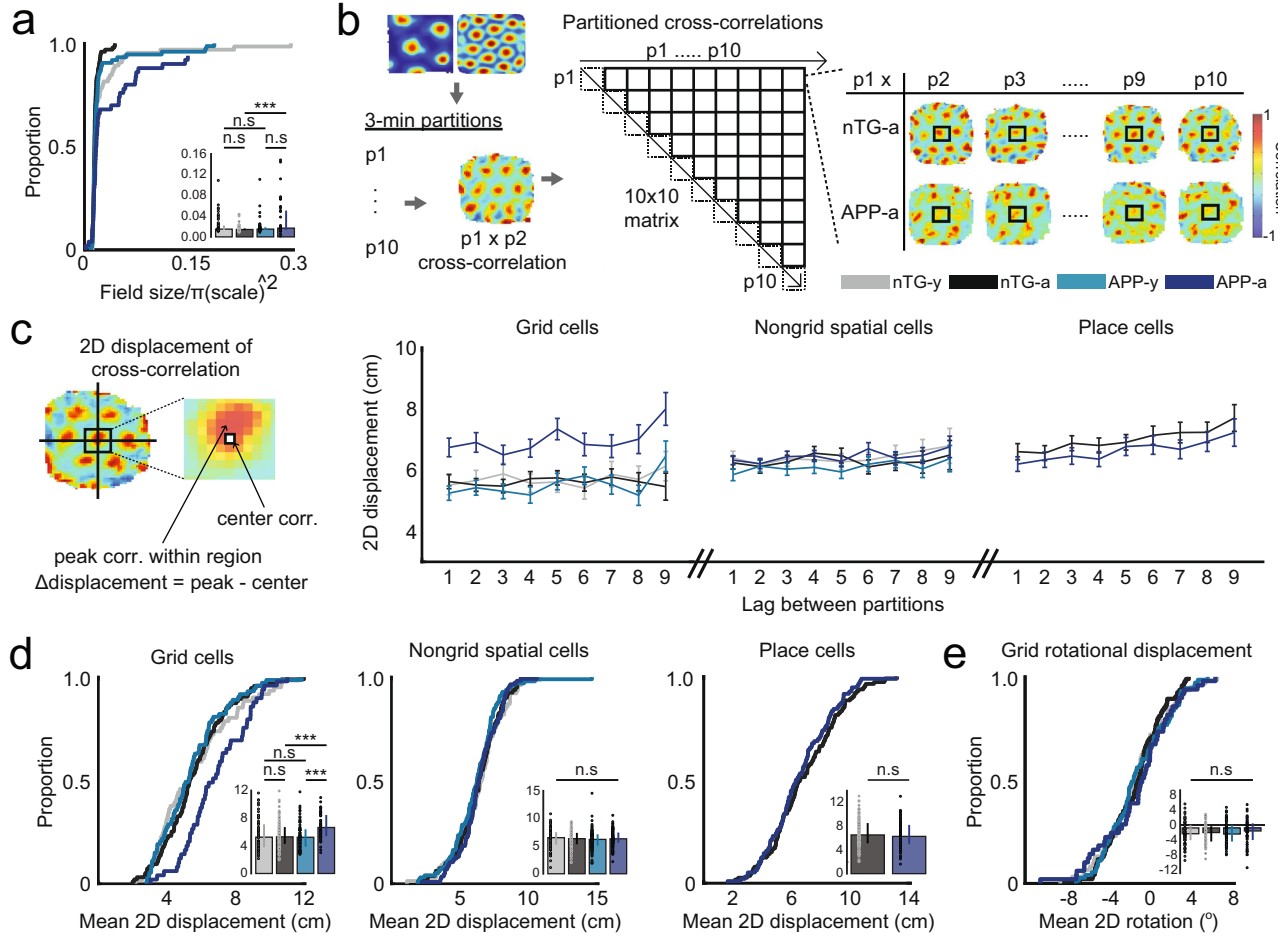

**Fig. 3 Grid cells in adult APP mice are spatially unstable. a** Normalized firing field size of grid cells (nTG-y–nTG-a: $P = 0.076$; nTG-y–APP-y: $P = 0.76$; APP-y–APP-a: $P = 0.52$; nTG-a–APP-a: $P = 0.0025$) between groups (cells, $n = 64$ nTG-y; $n = 99$ nTG-a; $n = 74$ APP-y; $n = 50$ APP-a). **b** Each grid cell recording was split into 10 3 min partitions. Two-dimensional spatial cross-correlations were computed across all partition pairs. Example cross-correlations of the first partition to subsequent partitions in two grid cells recorded from a nTG and an APP mouse are shown. **c** (Left) The two-dimensional spatial displacement was calculated as the distance between the peak correlation pixel and the center pixel of the cross-correlation. Note that this analysis makes no conclusions about the magnitude of the peak correlation pixel, and strictly assesses the shift of said peak value. (Right) Two-dimensional displacement of grid cells, non-grid spatially tuned cells and place cells as a function of lag between partitions. Dots indicate mean values and error bars indicate SEM. **d** Two-dimensional displacement of grid cells (nTG-y–nTG-a: $P = 0.76$; nTG-y–APP-y: $P = 0.76$; APP-y–APP-a: $P = 9.1 \times 10^{-4}$; nTG-a–APP-a: $P = 3.4 \times 10^{-4}$), non-grid spatially tuned cells (nTG-y–nTG-a: $P = 0.74$; nTG-y–APP-y: $P = 0.24$; APP-y–APP-a: $P = 0.19$; nTG-a–APP-a: $P = 0.87$), and place cells (nTG-a–APP-a: $P = 0.37$) between groups (grid cells, $n = 61$ nTG-y; $n = 95$ nTG-a; $n = 73$ APP-y; $n = 49$ APP-a; non-grid spatially tuned cells, $n = 77$ nTG-y; $n = 80$ nTG-a; $n = 98$ APP-y; $n = 115$ APP-a; place cells, $n = 114$ nTG-a; $n = 96$ APP-a). **e** Two-dimensional rotational displacement of one grid cell partition relative to another in the cross-correlation (nTG-y–nTG-a: $P = 0.15$; nTG-y–APP-y: $P = 0.87$; APP-y–APP-a: $P = 0.73$; nTG-a–APP-a: $P = 0.15$) between groups (grid cells, $n = 61$ nTG-y; $n = 95$ nTG-a; $n = 73$ APP-y; $n = 49$ APP-a). nTG-y non-transgenic young, nTG-a non-transgenic adult, APP-y APP young, APP-a APP adult. Wilcoxon rank-sum tests (two-sided) corrected for multiple comparisons using a Bonferroni-Holm correction were applied to analyze the data in panels **a**, **d**, and **e**. Data in bar graphs are presented as median values ± 25th and 75th percentiles; ***$P < 0.001$; n.s not significant. Source data are provided as a Source Data file.

(ANOVA, genotype main effect: $P = 0.0038$; genotype main effect: $P = 0.0084$; interaction effect: $P = 0.013$, Supplementary Fig. 12). Pairwise comparisons using Tukey's Test revealed greater spatial instability in APP-a grid cells, but not in APP-a non-grid spatially tuned cells or place cells (APP-a grid cells vs. nTG-a grid cells: $P = 0.0064$; APP-a non-grid cells vs. nTG-a non-grid cells: $P = 1$; APP-a place cells vs. nTG-a place cells: $P = 0.99$). Instability persisted in adult APP grid cells when partition lengths were extended from 3 min to 5, 6, and 10 min (Supplementary Fig. 13). Consistent with previous literature[30,31], positional coverage and running speeds were higher in adult APP mice, suggesting that greater instability was not biased by insufficient exploration of the open field environment (Supplementary Fig. 14).

**Grid cells in APP mice have reduced spike-time synchrony with interneurons and head-direction cells.** Given that inhibition constitutes a major input for grid cell generation[32,33], we analyzed the firing properties of interneurons in APP mice. Across age, mean firing rates became elevated in adult APP mice (ANOVA, genotype main effect: $P = 0.0038$; genotype main effect: $P = 0.0084$; interaction effect: $P = 0.013$, Supplementary Figs. 15, 16), alluding to possible changes in inhibitory networks within the MEC. In particular, we noted that a significant proportion of interneurons in young and adult APP mice had slower theta rhythmicity and reduced theta power, suggesting a potential early impairment in spike-timing dynamics between grid cells and interneurons preceding the loss of grid cell spatial periodicity (Supplementary Figs. 15, 16). By computing spike-time cross-

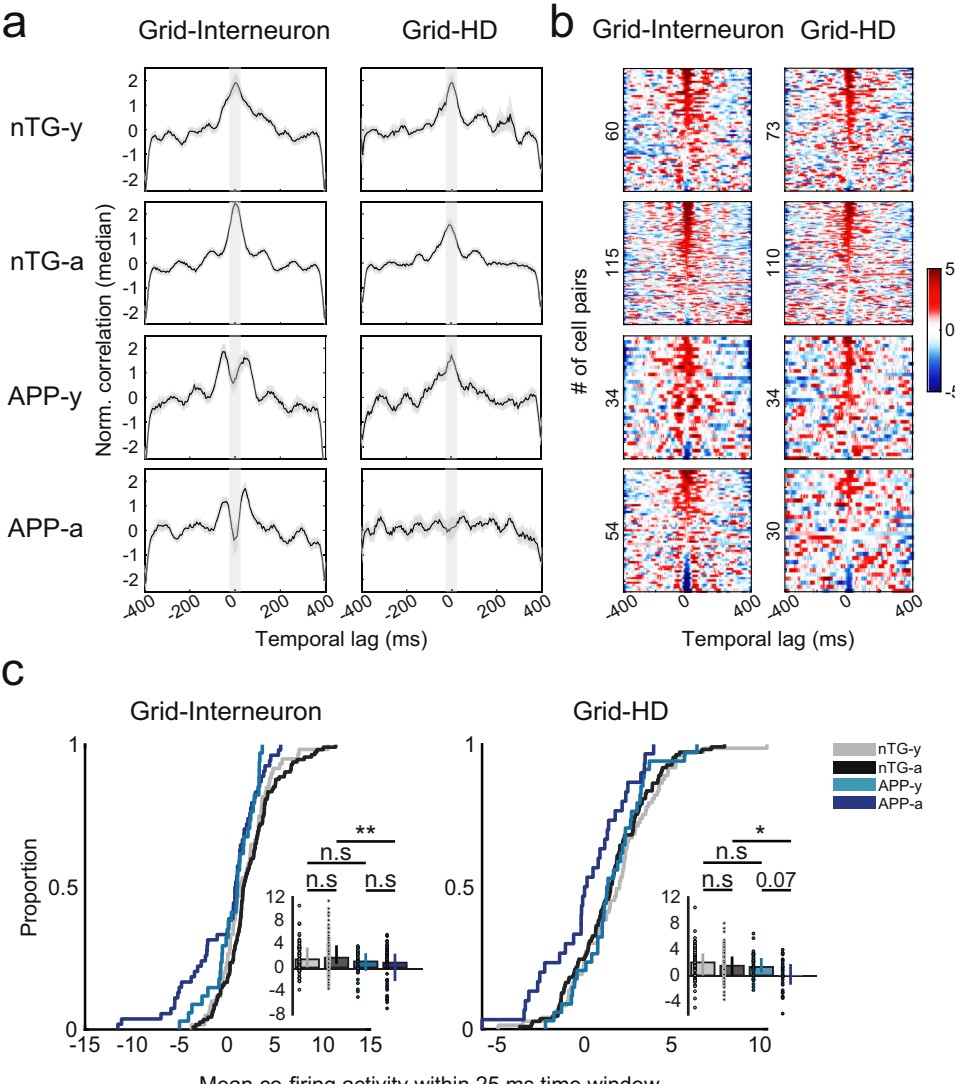

**Fig. 4 Reduced spike-time synchrony in grid cell-interneuron and grid cell-head-direction cell pairs. a** Spike-time cross-correlations between grid cell-interneuron pairs (left) and grid cell-head-direction cell pairs (right) for all experimental groups (rows). Each panel displays the normalized correlation (by median) by time lag (in ms). Black curves indicate median values and gray contours indicate median absolute deviation. Lighter gray columns indicate the 25 ms time window in each lag direction from 0 ms. **b** Color-coded raster plots show the magnitude of co-activity within a 400 ms time window. Y-axes are sorted in descending order by cell pairs with the maximum co-activity within a 25 ms time window, and numbers indicate the number of cell pairs in each experimental group. **c** The mean co-activity within a 25 ms time window for grid-interneurons pairs (nTG-y-nTG-a: $P = 0.30$; nTG-y-APP-y: $P = 0.13$; APP-y-APP-a: $P = 0.51$; nTG-a-APP-a: $P = 0.0018$), and grid-head-direction pairs (nTG-y-nTG-a: $P = 0.33$; nTG-y-APP-y: $P = 0.56$; APP-y-APP-a: $P = 0.078$; nTG-a-APP-a: $P = 0.038$) between groups (grid-interneuron pairs, $n = 60$ nTG-y; $n = 115$ nTG-a; $n = 34$ APP-y; $n = 54$ APP-a; grid-head-direction pairs, $n = 73$ nTG-y; $n = 110$ nTG-a; $n = 34$ APP-y; $n = 30$ APP-a). nTG-y non-transgenic young, nTG-a non-transgenic adult, APP-y APP young, APP-a APP adult. Wilcoxon rank-sum tests (two-sided) corrected for multiple comparisons using a Bonferroni-Holm correction were applied to analyze the data in panel **c**. Data in bar graphs are presented as median values ± 25th and 75th percentiles; *$P < 0.05$, **$P < 0.01$; n.s not significant. Source data are provided as a Source Data file.

correlations between simultaneously recorded MEC cells, we observed that synchrony between grid cells and interneurons were qualitatively reduced in young APP mice in comparison to nTG mice (Fig. 4a, b). In fact, young APP grid cells and interneurons appeared anti-synchronous at a temporal lag of ~25 ms, suggesting the start of early impairment of the grid cells' ability to temporally integrate inhibitory signals. Surprisingly, the same reduction in synchrony was also qualitatively observed between grid cells and head-direction cells, which appeared to worsen across age (Fig. 4a, b).

A two-way ANOVA was conducted to determine the effects of age and genotype on the mean co-activity within a 25 ms time window for grid cell-interneuron and grid cell-head-direction cell pairs (Supplementary Fig. 17, Fig. 4c). There was no significant interaction effect in either group (ANOVA: grid-interneuron interaction effect: $P = 0.091$; grid-head-direction interaction effect: $P = 0.083$ Supplementary Fig. 17, Fig. 4c), confirming the absence of an age-dependent reduction in synchrony. However, there was a significant main effect of genotype in both groups, indicating that grid cell-interneuron and grid cell-head-direction cell synchrony were impaired overall in both young and adult APP mice (ANOVA: grid-interneuron genotype main effect: $P = 1.3 \times 10^{-5}$; grid-head-direction genotype main effect: $P = 0.012$, Supplementary Fig. 17). In support of this view,

synchrony was significantly lower in adult APP mice compared to adult nTG mice, and was unaffected compared to young APP mice (Fig. 4c). However, the lack of statistical significance between young APP and young nTG mice implies that this reduction may be milder during the earliest stages of pathology (Fig. 4c). These findings are noteworthy for two reasons. First, given the importance of inhibitory and head-direction information for grid cell spatial firing[32–34], these results suggest that disrupted grid cell spatial periodicity across age in APP mice (Fig. 1) arises in part due to the temporal decoupling of grid cells from inhibitory and head-direction inputs within the local MEC network. Second, this decoupling starts (albeit mildly) at an age when the grid pattern is still intact, implying that grid cell coding is affected by early pathology preceding the complete loss of spatial periodicity.

**Grid cell impairments in APP mice correlate with worsened path integration performance.** Prior work has shown that APP mice exhibit spatial memory deficits on the Morris water maze and the radial arm maze by as early as 3–4 months of age[21,30]. Given the proposed role of grid cells in supporting path integration[17,35], we hypothesized that APP mice would also experience spatial memory deficits related to path integration. To test this hypothesis, we conducted a path integration task to assess the animals' ability to return directly to their refuge after finding a food pellet in an open field in complete darkness with an independent, non-implanted cohort of APP and nTG mice (n mice = 12 APP-y, 9 APP-a, 10 nTG-y, 8 nTG-a; Fig. 5a, Supplementary Fig. 18a). APP and nTG mice demonstrated a similar inclination to return to the refuge prior to consumption of the pellet (Supplementary Fig. 18b–d). However, we observed that APP mice were impaired in all measures of path integration ability relative to age-matched controls, with the greatest behavioral deficits in adult APP mice. In particular, the probability of arriving at the refuge during the initial wall contact decreased in APP mice across age (APP-a: 29%, APP-y: 38%, nTG-y: 58%, nTG-a: 57%; Fig. 5b, c), suggesting that they had greater difficulty in estimating their position relative to the refuge. In further support of this possibility, adult APP mice exhibited increased error in both their initial heading direction and the angular difference between the refuge and the first wall encountered during the return trajectory (Fig. 5d, e). With regards to overall navigational efficiency, adult APP mice travelled longer distances to return to the refuge and exhibited greater thigmotaxis by spending a larger proportion of the return path along the periphery of the environment (Fig. 5d, e). All groups showed improved performance when visual cues were made available (Supplementary Fig. 19), though APP mice remained impaired across all measures of task performance which worsened with age (Supplementary Fig. 20). Together, these results show that path integration abilities decline with age in APP mice, closely mirroring the timecourse of the spatial coding deficits observed in the grid cell network.

Lastly, we characterized which molecular changes could explain these early network alterations in the entorhinal-hippocampal circuit. A recent meta-analysis confirmed that synapse loss and changes in synaptic marker expression are major events in AD pathogenesis[36]. Likewise, an altered synaptic function could also affect circuit function such as grid cell coding, which is known to require both excitatory and inhibitory drive[28,32,33]. For these reasons, we carried out immunoautoradiography in the MEC and CA1 to visualize and quantify the expression of synaptic markers that include VGLUT1, VGLUT3, VAChT, VGAT, and NR1 (Supplementary Fig. 21). VGLUT1, VGLUT3 VAChT, and VGAT are neurotransmitter transporters

whereas NR1 is a subunit of NMDA receptors that was previously shown to be necessary for both grid cell integrity and path integration ability[35] (detailed marker descriptions in Methods, Immunoautoradiographic labelling of synaptic markers).

To interpret the most robust pathological changes, we ran linear mixed models to pinpoint which marker expression levels were most affected by early Aβ pathology. Out of the ten experimental groups, two cases were significantly modulated by the effect of the subject's genotype: VGLUT3 in the MEC and VGLUT1 in CA1 (VGLUT3 in MEC, genotype effect: $P < 0.01$; VGLUT1 in CA1, genotype effect: $P < 0.01$; Supplementary Fig. 22). We observed an increase of VGLUT3 in both young and aged APP mice, indicating that CCK-positive interneurons are exerting greater influence in inhibitory circuits within the MEC (Supplementary Figs. 23, 24). However, VGAT levels were not significantly different (Supplementary Figs. 21, 22), suggesting that early Aβ pathology targets a specific inhibitory circuit while sparing overall inhibitory drive. Taken together with our spike-time cross-correlation analysis (Fig. 4), these findings pinpoint inhibitory mechanisms as one of the earliest network changes in the MEC. An increase of VGLUT1 was also detected in CA1 of young APP mice that stayed elevated across age (Supplementary Figs. 22, 23). This finding explains the higher mean firing rate of adult APP place cells (Fig. 2d), and supports existing evidence showing that hyperexcitability is a major pathological symptom of AD[37]. Taken together, these results provide an in-depth overview of the early network changes in the MEC-hippocampal circuit susceptible to Aβ pathology at the molecular, physiological, and behavioral levels.

## Discussion

To identify the impact of Aβ pathology on neural coding in the MEC-hippocampal circuit, we obtained single-unit recordings during the initial stages of disease in an APP mouse model of familial AD. These data revealed a disruption in entorhinal grid cell coding when initial Aβ fibrils are detected. In contrast, the spatial tuning of other functional cell types in the MEC and region CA1 of the hippocampus was preserved. Theta power and modulation of theta by running speed remained intact in adult APP mice, yet grid cells exhibited reduced theta rhythmicity and spatial stability. Grid cells in young APP mice appeared to be decoupled from interneurons and head-direction cells, which worsened across age. These changes in grid cell coding corresponded with impaired performance of adult APP mice in a path integration task. Together, these results reveal that early Aβ pathology targets the entorhinal grid cell network within the MEC-hippocampal circuit.

Our results address several possible circuit-level explanations that could underlie reduced grid cell coding in APP mice. Prior studies have shown that inputs from the anterior thalamic nuclei (ATN), the dorsal hippocampus, and the medial septum are each independently necessary for normal grid cell function. Entorhinal head-direction cells, which are dependent on direct and indirect inputs from the ATN[34], were preserved in adult APP mice, suggesting that projections from the ATN were intact. Place cells in the dorsal hippocampus remained spatially selective, stable, and had high firing rates, indicating that reduced feedback from the hippocampus cannot explain grid cell deficits in adult APP mice[28]. Finally, theta power and speed modulation of theta were preserved in adult APP mice, suggesting that medial septal theta-generating inputs to the MEC are conserved[25,26]. Nevertheless, our findings could still indicate a subtle impairment of basal forebrain inputs that innervate the grid cell network; one candidate could be decreased septal cholinergic inputs[38], as a selective loss of basal forebrain cholinergic neurons in the nucleus basalis of Meynert is observed in familial AD patients[39,40].

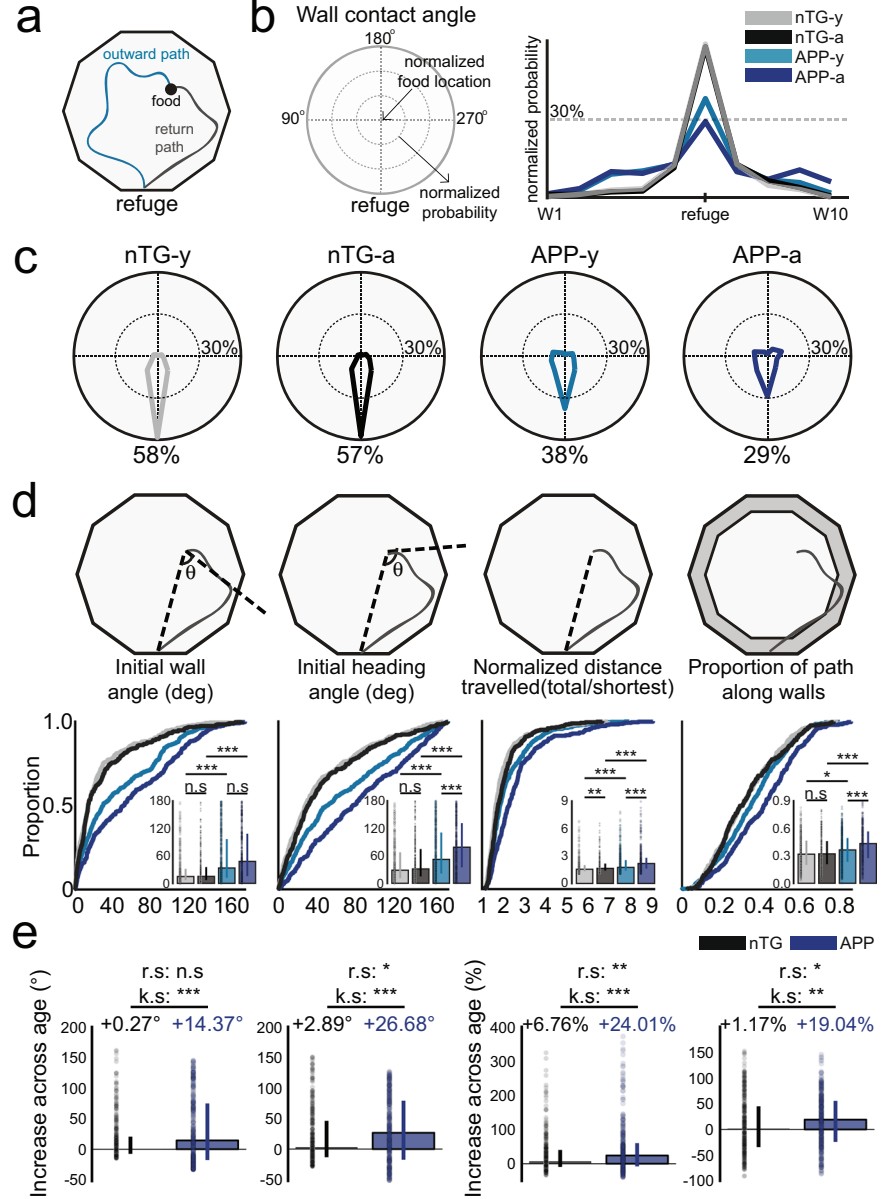

**Fig. 5 Path integration is impaired in APP mice and worsens with age. a** Food-foraging task in darkness where mice forage for a food pellet and navigate back to the refuge for consumption. **b** Likelihood of arriving at each wall (W1–W10) during the initial wall encounter of the return path between groups. **c** Probability of reaching the refuge during the initial wall encounter of the return path between groups. **d** Initial wall angle (nTG-y-nTG-a: $P = 0.50$; nTG-y-APP-y: $P = 6.3 \times 10^{-15}$; APP-y-APP-a: $P = 0.11$; nTG-a-APP-a: $P = 5.6 \times 10^{-15}$), initial heading angle (nTG-y-nTG-a: $P = 0.22$; nTG-y-APP-y: $P = 3.1 \times 10^{-10}$; APP-y-APP-a: $P = 1.9 \times 10^{-5}$; nTG-a-APP-a: $P = 9.5 \times 10^{-17}$), normalized distance travelled (nTG-y-nTG-a: $P = 0.0083$; nTG-y-APP-y: $P = 1.2 \times 10^{-6}$; APP-y-APP-a: $P = 1.2 \times 10^{-6}$; nTG-a-APP-a: $P = 3.9 \times 10^{-11}$), and thigmotaxis (nTG-y-nTG-a: $P = 0.74$; nTG-y-APP-y: $P = 0.011$; APP-y-APP-a: $P = 7.1 \times 10^{-5}$; nTG-a-APP-a: $P = 1.6 \times 10^{-7}$) between groups (trials, $n = 377$ nTG-y; $n = 307$ nTG-a; $n = 500$ APP-y; $n = 311$ APP-a). **e** Increase (degrees) of initial wall angle (nTG-a-APP-a: r.s $P = 0.42$; k.s $P = 2.3 \times 10^{-11}$) and initial heading angle (nTG-a-APP-a: r.s $P = 0.019$; k.s $P = 8.1 \times 10^{-6}$) across age, as well as increase (%) of normalized distance travelled (nTG-a-APP-a: r.s $P = 0.0029$; k.s $P = 8.5 \times 10^{-4}$) and thigmotaxis (nTG-a-APP-a: r.s $P = 0.014$; k.s $P = 0.0013$) across age (trials, $n = 307$ nTG; $n = 311$ APP). Median value changes across age for each group are indicated above bars. nTG-y non-transgenic young, nTG-a non-transgenic adult, APP-y APP young, APP-a APP adult. Wilcoxon rank-sum tests (two-sided) corrected for multiple comparisons using a Bonferroni-Holm correction were applied to analyze the data in panel **d**. Wilcoxon rank-sum tests (r.s) (two-sided) and two-sample Kolmogorov–Smirnov (k.s) tests (two-sided) were applied to analyze the data in panel **e**. Data in bar graphs are presented as median values ± 25th and 75th percentiles; *$P < 0.05$, **$P < 0.01$, ***$P < 0.001$; n.s not significant. Source data are provided as a Source Data file.

We also observed that grid cells recorded in APP mice had reduced spike-timing synchrony with interneurons and head-direction cells. This is interesting because interneuron means firing rates and head-direction cell directional selectivity were not lower in APP mice. These results are particularly noteworthy because grid cells require excitatory, inhibitory and head-direction inputs[28,32–34]. While we did not observe any obvious impairments in those systems specifically in terms of basic firing properties, their temporal integration within the grid cell network could potentially disrupt spatial coding. The temporal decoupling of grid cells from local inputs therefore provides a network-level explanation for the reduced grid cell spatial periodicity and

stability observed in adult APP mice. On that note, we could not analyze grid cell-place cell synchrony, given that we did not record from the MEC and CA1 simultaneously. This decoupling is also weakly present in young APP mice (as suggested by a two-way ANOVA, but insignificant via direct non-parametric testing), alluding to the possibility that grid cell coding is impaired prior to the complete loss of spatial periodicity. It could be that the temporal decoupling is progressive and worsens with age, but we cannot directly confirm this idea with the current data and the lack of significant effect size in young APP mice. Lastly, the cross-correlation analyses should not be used to draw conclusions regarding the amount of inhibition or excitation integrated by grid cells, which require other experiments to validate.

We also characterized whether local network-level changes can be accounted for at the molecular level. Based on our quantification of synaptic markers, VGLUT3 levels were pathologically elevated in the MEC of APP mice. These findings pinpoint specific inhibitory mechanisms as one of the earliest network changes in the MEC, as VGAT levels indicative of global inhibition were unaltered. However, these findings are hard to relate to the observed grid cell impairment. On the other hand, an increase of VGLUT1 was also detected in CA1 of young APP mice that stayed elevated across age. This finding explains the higher mean firing rates of adult APP place cells and is consistent with the hypothesis that hyperexcitability is a major pathological symptom of AD[37]. We applied a linear mixed model and only considered marker expression levels that were significantly predicted by the subject's genotype. By doing so, a more subtle effect amongst other markers might have been deliberately missed. For instance, there was a slightly lower NR1 expression in adult APP mice relative to adult nTG mice, and it is known that this NMDA receptor subunit is necessary for grid cell firing[35].

Our results suggest that grid cells contribute to path integration, and possibly other forms of spatial memory. Young APP mice were modestly impaired in our path integration task despite an intact grid cell spatial periodicity. In parallel, prior work has shown spatial memory impairments in this APP mouse line at the same age in the radial arm and Morris water mazes[21,30]. Early disruption of the spike-timing relationship between grid cells and other MEC cell types in young APP mice could potentially underlie these behavioral impairments. Likewise, reduced grid cell spatial periodicity and stability could explain the more pronounced path integration impairments in adult APP mice. Our behavioral data showed that in addition to increased travel distance and angular errors in the return path, adult APP mice spent more time along the environment periphery. This suggests that adult APP mice could not plan effective routes back to the refuge and instead adopted a thigmotaxic strategy. The severity of grid cell deficits paralleling the worsened behavioral performance provides compelling evidence to suggest that this microcircuit is linked to path integration and perhaps other forms of spatial memory. There are likely other undetermined factors that explain the spatial memory impairment observed in these mice at a young age, but our findings are consistent with the current understanding of how grid cells are necessary for proper path integration function[17,35].

Of particular importance, our results reveal that Aβ-mediated perturbations at the synaptic level do not uniformly impact neural computations. APP mice express soluble Aβ oligomers throughout the entorhinal cortex and exhibit reduced density of presynaptic terminals and neuronal loss across all entorhinal layers by 6 months of age[20], yet only the grid cell subnetwork within the MEC was disrupted. Similar and widespread changes are observed throughout the hippocampus, but the spatial coding of place cells was not disrupted.

Place cells have also been recorded in other APP-related mouse models. The spatial tuning of place cells in the Tg2576 APP

mouse model remained intact at the earliest incidence of Aβ plaques[27], similar to our results. However, differences in place cell physiology have also been reported. In the 3xTg triple transgenic mouse model displaying both APP and tau pathology, place cells exhibited spatial instability on a linear track preceding the detection of plaques which seems to be in conflict with our findings[41]. One explanation for this difference is environmental influence. As the mouse's freedom of movement is constrained on a one-dimensional track, the direction of instability is spatially restricted. In contrast, instability in an open field can occur in 360 degrees and averaging these directional shifts over time may ultimately cancel out to give the impression that APP place cells are stable. It could be that a linear track is more sensitive at detecting precise changes in place cell stability that may not meaningfully impact overall spatial coding in a two-dimensional environment. Alternatively, this difference could be due to the presence of tau pathology in 3xTg mice. In a different study involving a chimeric APP mouse model where the onset of APP expression could be controlled, place cell stability was also impaired on a linear track[42]. However, this disruption took place 9.5 months following APP expression, a pathogenic timepoint that is much later than ours which could explain their results. Lastly, a study reported that grid cells and place cells were disrupted in an APP knock-in mouse model[43]. Despite already having moderate levels of plaque formation throughout the brain, these young APP knock-in mice still did not show any impaired place cell coding, which is consistent with our findings.

It is certain that inherent differences within mouse models may contribute to variability between results[41,42], but so can the experimental design. Our place cell results are best compared to those recorded from Tg2576 mice[27] because recordings were done in an open field during the earliest detection of amyloid plaques. From this perspective, our results are consistent with what is currently known about Aβ pathology and place cell coding. To this growing body of knowledge, we show that impairments in grid cell firing emerge prior to place cell disruption. Importantly, both extracellular and intracellular Aβ-related processes may be pathogenic drivers of the reported network changes and should be further investigated. Despite the popular belief that extracellular Aβ initiates many aspects of pathology, there is a wide body of evidence showing that intracellular Aβ does the same[44–47]. Alternatively, functional magnetic resonance imaging (fMRI) has revealed that the lateral entorhinal cortex (LEC) could be the first region affected in early AD[48]. The LEC is an important node in the entorhinal-hippocampal circuit and has also been studied in APP mouse models. In PDAPP mice, the location of amyloid deposits in the dentate gyrus greatly coincided with the termination of afferent projections from the LEC[49]. In terms of single-unit physiology, a report showed that cells in Tg2576 mice displayed hyperactivity in the LEC by as early as 3 months of age[50]. Physiological changes in the LEC as a result of Aβ may precede the reported grid cell impairments and merit further investigation.

There is concern regarding the use of transgenic APP mice such as the J20 model that overexpresses non-physiological Aβ given that recent APP knock-in mice express pathological profiles that are more faithful of AD pathogenesis. Caution should be exerted when relating the conclusions of this study to human AD populations. Nevertheless, we took advantage of the robust phenotypic nature of J20 mice to identify the specific parts of the MEC-hippocampal spatial coding circuit most impacted by early APP pathology. Importantly, AD is also a multifaceted neurodegenerative disease marked by pathological markers other than Aβ, such as widespread neurofibrillary tangles consisting of the hyperphosphorylated-tau protein. Therefore, the results of our study do not provide a complete overview of grid cell dysfunction

in AD. Yet, it is interesting to note that prior work using a tau transgenic mouse line has shown that grid cell spatial coding is preserved when tau is initially restricted to axonal and somato-dendritic compartments, but is subsequently impaired once tau has accumulated extensively in entorhinal cell bodies[51].

Reports on multi-study validation of data-driven disease progression in human AD patients[52,53] predict that cohorts of familial AD and *APOE-ε4*-positive subjects exhibit cerebrospinal fluid biomarkers in a distinct sequence: amyloid-β1–42, phosphorylated tau, and then total tau. However, in the broader AD population, total tau and phosphorylated tau are found to be earlier biomarkers than Aβ. The combined findings that early Aβ and advanced tau pathologies each independently target the grid cell network highlight the vulnerability of this entorhinal sub-network and raise the possibility that spatial memory deficits in AD are linked directly to grid cell integrity. Indeed, functional imaging in young adults at genetic risk of AD (*APOE-ε4* carriers) revealed a reduced grid-like hexa-symmetric signal in the MEC that correlated with spatial memory and path integration impairments[54–56]. These convergent lines of evidence support the viability of grid cell integrity and spatial navigation deficits as early AD markers[4], and as dependent variables to assess the efficacy of AD therapeutics.

## Methods

**Subjects**. J20 APP male mice (B6.Cg-Zbtb20 Tg(PDGFB-APPSwInd) 20Lms/2Mmjax) were obtained from The Jackson Laboratory (MMRRC stock #34836) and bred with female C57/BL6/j mice. Mice were individually housed on a 12 h light/dark cycle and underwent experiments during the light cycle. The housing room conditions of the mice were maintained at 20–22 degrees Celsius and 21–30% humidity. All experimental procedures were performed in accordance with McGill University, Douglas Hospital Research Centre Animal Use and Care Committee (protocol #2015-7725) and Canadian Institutes of Health Research guidelines.

In J20 mice, layers 2, 3, and 5 of the MEC undergo progressive neuronal loss and by 7.5 months of age, all layers experience a combined loss of 16.3% in comparison to age-matched controls[18]. The entorhinal cortex as a whole exhibits a reduced density of presynaptic terminals (quantified by synaptophysin-immunoreactivity) by 7 months of age[20]. Similarly, by 6 months of age, region CA1 of the hippocampus in APP mice exhibits a 10%+ loss of neurons compared to age-matched controls[21]. Synapse loss is observed as early as 3 months of age in CA1, confirmed both by synaptic marker-immunoreactivity and electron microscopy[22]. In addition to these processes, the complement-dependent pathway and microglia undergo aberrant upregulation that is dependent on soluble Aβ oligomeric levels in the hippocampus[22]. Furthermore, gliosis (activated astrocytes) and neuroinflammation (activated microglia) become elevated by 6 months of age in the hippocampus[21]. Lastly, in vitro slice electrophysiology experiments revealed that both basal synaptic transmission recorded in CA1 and long-term potentiation in the Schaffer collateral–CA1 synapse are impaired by 3 months of age[23]. To examine the impact of these Aβ-mediated changes on neural coding circuits during these early stages of Aβ pathology, we focused on studying APP mice between 3–7 months of age.

Single-unit recording data in the MEC were collected from 68 APP mice and littermates with negative transgene expression across four experimental groups: young APP mice (3–4.5 months of age), adult APP mice (4.5–7 months of age), young non-transgenic (nTG) mice (3–4.5 months of age), adult nTG mice (4.5–7 months of age). Thirty-one males and 37 females were used. Ten animals fell into multiple age groups. The male/female ratios were 6:5, 16:16, 9:5, and 11:10 for young APP, adult APP, young nTG, and adult nTG mice, respectively. Single-unit recording data in region CA1 of the hippocampus were collected from six adult APP mice (3:3 male/female ratio) and six adult nTG mice (2:4 male/female ratio).

A separate, non-implanted cohort of APP and nTG mice were tested in the path integration behavior task. Mice were separated into the same four experimental groups defined above. The male/female ratios were 6:6, 6:3, 5:5, and 4:4 for young APP, adult APP, young nTG, and adult nTG mice, respectively.

**Surgery**. On the day of surgery, mice were anesthetized with isoflurane (0.5–3% in $O_2$) and administered carprofen (0.01 ml/g) subcutaneously. For each mouse, three anchor screws were secured to the skull and a ground wire was positioned either above the cerebellum at midline position or the left visual cortex. A 'versadrive' containing four independently movable tetrodes (Axona, Inc) was implanted on top of the right MEC at the following stereotaxic coordinates: 3.4 mm lateral to the midline, 0.25–0.40 mm anterior to the transverse sinus. For hippocampal implants, the versadrive was implanted on top of the right CA1 at the following stereotaxic coordinates: 1.5 mm lateral to the midline, 1.9 mm posterior from bregma. Tetrodes were gold-plated to lower impedances to 150–250 kΩ at 1 kHz prior to

surgery. The versadrive was angled at eight degrees in the posterior direction for MEC implants and was not angled for CA1 implants. Following placement, the versadrive was secured in place using Kwik-Sil (to prevent exposure of the brain) and dental acrylic (to secure the versadrive to the skull and anchor screws). The ground wire was soldered to the implant, and tetrodes were lowered 1.0 mm and 0.5 mm from the dorsal surface for the MEC and CA1, respectively. All surgical procedures were performed in accordance with McGill University, Douglas Hospital Research Centre Animal Use and Care Committee (protocol #2015-7725) and Canadian Institutes of Health Research guidelines.

**Neural recordings**. Three days post-surgery, mice were placed on water restriction and maintained at 85% of their ad libidum weight for the duration of experiments. Mice were tested in six different open field environments. The majority of MEC recordings were done in a 75 × 75 cm box (1109 recordings), but a number of them also took place in a ten-sided maze with a 63.8 diameter (9 recordings), a 50 × 50 cm box (121 recordings), a 84 × 84 cm box (23 recordings), a 90 × 90 cm box (1 recording), and a 100 × 100 cm box (58 recordings). All CA1 recordings were done in the same 75 × 75 cm box. As mice explored their environments, water droplets were randomly scattered throughout to motivate the subjects to adequately sample the entire open field. Once mice reliably provided good trajectory coverage, tetrodes were turned quickly until theta rhythmic units were observed which indicated that the tetrodes had entered the MEC. Tetrodes were then advanced in increments of 25 microns to sample new putative MEC neurons, which was later confirmed by histology. For the CA1 cohort, sleep recordings were carried out prior to open field exploration to detect sharp wave and ripple activity. Once ripple amplitude was stable across days, tetrodes were no longer turned. Occasionally, tetrodes were either advanced or retracted depending on fluctuations in ripple amplitude and unit activity. In most cases for both MEC and CA1 recordings, neurons were not stable enough between recordings to reliably determine whether cells were re-sampled across days and thus we have included all cells recorded into our analysis.

To record spikes and local field potentials, versadrives were connected to a multichannel amplifier tethered to a digital Neuralynx (Bozeman, MT) recording system, and data were acquired with Cheetah 5.0 software (Neuralynx, Inc). Signals were amplified and band-pass filtered between 0.6 kHz and 6 kHz. Spike waveform thresholds were adjusted before commencing each recording and ranged between 35 and 140 μV depending on unit activity. Waveforms that crossed the threshold were digitized at 32 kHz and recorded across all four channels of the given tetrode. Local field potentials were recorded across all tetrodes.

**Histology**. Animals were anesthetized with isoflurane and perfused intracardially using saline, followed by 4% paraformaldehyde. Animal heads were left in 4% paraformaldehyde for between 24 and 72 h following perfusion, before brains were extracted. Brains were left to sink in a 30% sucrose solution, and then frozen and stored in a −80 °C freezer. Sagittal brain sections (40 μm) were sliced using a cryostat and Nissl-stained with a Cresyl violet solution. In cases where brain slices repeatedly came off the glass slides during Nissl-staining, slices were instead mounted using a fluorescent DAPI labeling mounting medium.

Tetrode tracks were characterized to be in either the superficial or deep layers based on the location of the track tip. Only data collected from tetrodes within the MEC were included in the analysis.

For hippocampal recordings, all tetrode tips that picked up single units were determined to be in region CA1 of the dorsal hippocampus. Tips from tetrodes located outside of CA1 did not pick up any single units.

**Spike sorting**. Single units were isolated 'offline' manually using Offline Sorter 2.8.8 (Plexon, Inc) individually for each recording session. Neurons were separated based on the peak amplitude and principal component measures of spike waveforms. Evaluation of the presence of biologically realistic interspike intervals, temporal autocorrelations, and cross-correlations was used to confirm single-unit isolation. The experimenter was blind to the age and genotype of the subjects and only well-separated clusters were included in the analysis.

**Position, direction, and velocity estimation**. For all electrophysiological recordings, positional data were acquired at 30 frames per second at 720 × 480 pixel resolution (4.9 pixels per cm) using a camera purchased from Neuralynx (Bozeman, MT). The camera was elevated at a height such that it fully captured all recording environment sizes used. The estimated position of the animal was calculated as the centroid of a group of red and green diodes positioned on the recording head stage. Head direction was calculated as the angle between the red and green diodes. Up to five lost samples due to occlusion of tracking LEDs, or reflections in the environment were replaced by linear interpolation for both position and directional data. Running velocity was calculated using a Kalman filter. Rate maps were constructed by calculating the occupancy-normalized firing rate for 3 × 3 cm bins of position data. Data were smoothed by a two-dimensional convolution with a pseudo-Gaussian kernel involving a three pixel (9 cm) standard deviation. To visualize the periodicity of grid fields, we computed the spatial autocorrelation of the smoothed rate maps using Pearson's product-moment correlation coefficient as described in Supplementary Fig. 3.

**Gridness score**. To quantify the spatial periodicity of MEC neurons, we calculated a 'gridness score' as described in Brandon et al.[25]. Briefly, this metric quantifies the hexagonal spatial periodicity in firing rate maps, while also accounting for elliptical eccentricity along with one of two mirror lines that exist in a hexagonal lattice structure. Distortion along one of the mirror lines was corrected after determining the major and minor axes of the grid based on the six fields closest to the central peak of the rate map autocorrelogram. The entire autocorrelogram was compressed along the major axis so that the major axis became equal to the minor axis. Large eccentricities (where the minor axis was less than half of the major axis) were not corrected. From the compressed autocorrelogram, we extracted a ring that encased the six peaks closest to the center peak but excluded the central peak to report periodicity between fields. We then calculated a rotational autocorrelation of this ring and observed the periodicity in paired pixel correlations across 180 degrees of rotation. The gridness score was computed as the difference between the lowest correlation observed at 60 or 120 degrees of rotation and the highest correlation observed at 30, 90, or 150 degrees of rotation. To ensure that our finding that grid cell reduction was not observed because of double-sampling grid cells across recording sessions, we made efforts to reduce putative double-sampling. Recordings of grid cells with cluster centroids within 0.2 mV on subsequent days were considered to be putative duplicate recordings, and the grid cell recording with the best separation index was chosen for statistics on gridness across groups in Supplementary Fig. 7. We used the full set of recordings for all other analyses.

**Directionality**. Polar histograms of firing rate by head direction were generated to visualize the pattern of spiking dependent upon the animal's direction. To construct the polar plots, the head direction was collected into bins of 6 degrees and the number of spikes in each bin was divided by the time spent facing that direction. The mean resultant length (MRL) of the polar plot was taken as a metric of head-direction selectivity.

**Cell selection**. We categorized each entorhinal neuron as a grid cell, head-direction cell, or non-grid spatially tuned cell. We performed a shuffling procedure to set significance criteria to determine grid cells and head-direction cells. Spike trains from each neuron recorded were randomly shifted in time by at least 30 seconds. We then calculated gridness and directionality measures. This process was repeated 50 times for each neuron, and the 99th percentile of the resulting distribution of scores was determined as the significance criteria for both measures. This results in a gridness threshold of 0.54 and a directionality threshold of 0.21 which we used to define grid cells and head-direction cells in our full dataset. Any cell recorded in the MEC which did not qualify as a grid cell but had a split-half correlation ≥0.6 was categorized as a non-grid spatially tuned cell. Putative interneurons in the MEC were selected by having a narrow waveform (<0.3 ms) and a mean firing rate of at least 0.5 Hz. Hippocampal neurons were classified as putative place cells if they had (1) a minimum mean firing rate of 0.1 Hz, (2) a maximum mean firing rate of 5.0 Hz, and (3) a spatial peak rate of greater than 5.0 Hz. Duplicate place cells sampled across recording sessions were removed for Figs. 2d and 3d.

**Spatial 2D displacement analysis**. To quantify noise in the two-dimensional (2D) phase of grid cells (and other cell types) on short timescales, we began by dividing the first 30 min of each recording into 10 epochs of three minutes each. For each epoch, we computed the resulting rate map. Next, for all pairwise comparisons of epoch rate maps, we computed the spatial cross-correlation between rate maps over a window of ±5 pixel (±15 cm) lags in both dimensions. The peak of this cross-correlogram captures the 2D translation necessary to best align the current pair of rate maps. Because the periodic nature of the grid pattern might lead to multiple local maxima in the cross-correlogram, we first computed the patch of correlation values nearest the center for which all contiguous correlation values were at least 50% of the maximum correlation value. We then chose the maximum correlation in this patch as our peak. The distance from the center (no difference in alignment) to this peak was computed as our measure of 2D phase-shift between these epochs. The average across all pairwise comparisons of epochs was then the final measure of 2D phase noise for that cell.

**Speed modulation of theta power and frequency**. Local field potential traces obtained from the MEC were referenced to a cortical reference electrode and downsampled to 500 Hz. Power between 1 and 15 Hz was calculated using a Morlett Wavelet with a 0.25 Hz bandwidth to obtain a power spectrum for each sample. Theta-by-Speed spectrograms were calculated as the power between 5 and 15 Hz divided by power in the delta band (2–4 Hz) across running speeds. The average Theta-by-Speed spectrogram is shown in Fig. 2e. To quantify speed modulation of theta power, the mean power between 7 and 12 Hz across speeds was extracted from each Theta-by-Speed spectrogram across speeds (Fig. 2e). To quantify speed modulation of theta frequency, the frequency of the peak power for each running speed was extracted from the Theta-by-Speed spectrogram (Fig. 2e). Analysis of covariance (ANCOVA) was performed on these extracted data.

**Single-cell temporal autocorrelations and intrinsic frequency**. The spike times of each cell were binned at 5 ms intervals and the temporal autocorrelation for the given spike train was computed. The obtained signal was smoothed by a Gaussian kernel with 2 bin standard deviation, zero-padded to $2^{13}$ samples and the power spectrum was calculated using the Chronux toolbox function MTSPECTRUMC from Matlab. The intrinsic frequency of a given cell was then taken as the frequency with the max power in the 6–12 Hz range.

**Cross-correlations and synchrony analysis**. To examine spiking synchrony, unbiased cross-correlations were computed between simultaneously recorded grid cells, head-direction cells, and putative interneurons with 5 ms temporal bins from a lag of −400 to 400 ms. The resulting cross-correlations were convolved with a 25 ms gaussian and normalized to their median absolute deviation for comparison.

**Path integration task**. Data were collected in a ten-sided maze (diameter = 63.8 cm) surrounded by black curtains. Steel bars were screwed into the walls of the testing room and hovered over the maze. A plastic base was positioned on these bars and acted as the ceiling for the maze. On this ceiling, an infrared camera purchased from Neuralynx was positioned and acquired positional data at 30 frames per second at 720 × 480 pixel resolution (6.13 pixels per cm). Black curtains were positioned on top and around this plastic base which draped over the maze and ensured a completely dark environment. Within the maze, 10 refuge enclosures connected to the open environment were closed off by top-down sliding doors that acted as walls. For all trials, the same refuge was used for each subject. When the door was slid open, the mouse could voluntarily enter or exit the refuge by their own volition. The height of these walls (and the entire maze throughout) was 27.6 cm.

Mice were placed on food restriction and maintained at 85% of their ad libidum weight throughout training and testing phases. In each trial, the mouse was kept in the same refuge enclosure separated from the open environment by the sliding door. The maze was operated in darkness via a pulley system which consisted of a rope fastened to the top of the sliding door. This rope extended outside of the curtains by passing through 2 clamps that were installed along the steel bars above the maze. This setup mimicked a pulley system where the experimenter could pull on the rope and open the sliding door while the curtains were draped over the maze. The handle of the rope end was twisted into a knot; at the start of each trial, the rope was pulled, and the knot was looped onto a third clamp fastened to a table post. Doing so kept the sliding door held up throughout the duration of each trial. At the end of a trial, the knot was lifted from the clamp which closed the door. This setup allowed the experimenter to quickly operate the door without needing to physically interact with the maze.

Once the mouse was let into the open environment, it had to forage for a randomly placed small food pellet and return to the refuge prior to consumption. These food pellets were the same kind as administered in the subjects' cages, but smaller in size weighing less than 0.2 g. Successful trials were defined as events where the mouse picked up the food pellet and navigated to the refuge before consumption. Failed trials were defined as events where the mouse failed to return to its refuge before consuming the pellet. Incomplete trials were defined as events where the mouse failed to retrieve the pellet before returning to its refuge.

Visual cues were setup along the walls of the environment to allow for increased allocentric-guided behavior in the light trials. The three visual cues used consisted of a triangle, square, and three stripes constructed using tape and were positioned on three almost-equally spaced walls (given that the environment is ten-sided, a cue couldn't be completely equally spaced from the other two). White noise played throughout all trials to account for potential auditory cues that may affect the mouse's return trajectory. Furthermore, the maze environment was wiped using Peroxyguard following every five consecutive trials to reduce the extent to which olfactory cues influenced behavior. In light trials, the room lighting was turned on and the curtains were pushed to the side. In dark trials, the room lighting was turned off and the curtains completely covered the arena. The mouse's movements were tracked using an overhead infrared camera, and the maze was lit using infrared light.

*Path integration behaviour timecourse*. Mice reached 85% of their ad libidum weight before experiments commenced. Mice first underwent a training phase where they achieved a minimum of eight successful trials out of ten total complete trials within a session in light conditions. Incomplete trials did not count as completed trials. Mice went through consecutive light training days until they reached the success criteria. During failed trials, the experimenter punished the mouse by holding it by the tail suspended in the air for ten seconds before placing it back into the refuge.

Following light training, mice then underwent five consecutive days of dark training. The same protocol as the light training applied to dark training. Mice were required to achieve a minimum of eight successful trials out of ten total complete trials within a session in any of the five days. All mice reported in the dataset achieved success criteria. Four mice that did not pass the training criteria were excluded from the analysis. These mice included two young nTG mice, one young APP mouse, and one aged APP mouse.

Following dark training, mice then underwent five consecutive days of light and dark testing. On days 1, 3 and 5, five light trials were conducted, followed by five dark trials. Incomplete trials counted as trials. This was repeated until the mouse achieved ten complete trials in each of the light and dark conditions. On days 2 and

4, the same protocol was applied, but the mouse started with five dark trials, followed by five light trials.

**Analysis of path integration behavior**. All path integration behavioral data were recorded at 30 frames per second. The positional coordinates of the mice for each trial were obtained using an open-source deep-learning tracker algorithm called DeepLabCut[57]. DeepLabCut was only used to quantify positional data in the path integration task and not for electrophysiological recordings. Custom Matlab scripts were used to analyze various behavioral parameters from the mice's positional data.

**Genotyping**. Tail samples were collected at weaning for genotyping, and just prior to brain perfusion for additional confirmation. DNA samples were extracted and amplified using the REDExtract-N-Amp™ Tissue PCR Kit (MilliporeSigma, XNAT-100RXN) and the primer sequence and PCR protocol provided by The Jackson Laboratory (MMRRC, 34836-JAX). Genotyping results were visualized using a QIAxcel instrument (Qiagen).

**Immunofluorescence**. Mice were anesthetized with Isoflurane (Baxter, FDG9623) and intracardially perfused with 0.05% heparin (Sandoz, 10750) in ice-cold saline followed up cold and filtered 4% paraformaldehyde that was freshly made from powder (MilliporeSigma, 158127-500 g). Extracted brains were cryopreserved in 30% sucrose (MilliporeSigma, S0389-1Kg), flash frozen in 2-methylbutane (Fisher Scientific, 03551-4), and kept at −80 °C until sliced on a cryostat (Leica, CM3050-S). Sagittal sections (40 μm) were collected on microscope slides for on-slide staining. Each slide had two positive controls (APP animals 18 months old) and at least one brain section from the remaining experimental groups (young APP, adult APP, young nTG, adult nTG). The same combinations of brain sections were used for both MEC and hippocampal staining. Sections that were too damaged were discarded. All slides were processed at the same time using the purified mouse monoclonal anti-beta-amyloid 1–16 antibody (6E10) (Biolegend, catalog number 803001) at a dilution of 1:500 for 30 minutes, along with the M.O.M.® Fluorescein Kit (Vector Laboratories, catalog number FMK-2201). Slides were mounted with DAPI containing Fluoromount-G (SouthernBiotech, catalog number 0100-20).

**Analysis of Immunofluorescence**. Images for each section were acquired within the same session at ×10 magnification with the same exposure settings (FITC: 250 ms, DAPI: 50 ms) on a slide scanner (Olympus, VS120) within one week of the immunofluorescence assay. The images were digitally processed using ImageJ[47]. ROIs were manually drawn for both MEC and the hippocampus and clear visually identifiable artifacts were removed from ROIs. Rolling ball background subtraction (70 μm radius) was applied to every image. ROI areas were measured, and fluorescence intensity was extracted. A threshold was set for analysis to capture the plaque fluorescence signal (6000 a.u.) based on beta-amyloid plaques observed in positive control animals. Using RStudio (RStudio Team 2016), the sections were grouped by structure: MEC or hippocampus. For each animal, the normalized fluorescence was calculated as the total fluorescence divided by the total area.

**Immunoautoradiographic labelling of synaptic markers**. Immunoautoradiography experiments were performed on fresh frozen mouse brain sections (10 μm) as described previously[58,59]. Brain slices were taken at the level of the MEC (bregma 2.76 to 3.90) and the hippocampus (bregma −1.0 to −2.0). Slices were incubated overnight at 4° with rabbit polyclonal antiserum specific of VGLUT1 (dilution 1:10,000), VGLUT3 (dilution 1:20,000, Synaptic Systems, catalog number 135203, Göttingen Germany), VGAT (dilution 1:10,000, Synaptic Systems, catalog number 131002, Göttingen Germany), VAChT (dilution 1:10,000, Synaptic Systems, catalog number 139103, Göttingen Germany), NR1 (dilution 1:10,000, Synaptic Systems, catalog number 114103, Göttingen Germany) and then with anti-rabbit [125I]-IgG (PerkinElmer, 0.25 μCi/ml final dilution) for 2 hr at 4°. Sections were then washed in PBS, rapidly rinsed in water, dried, and exposed to x-ray films (Biomax MR, Kodak) for 5 days. Standard radioactive microscales were exposed to each film to ensure that labeling densities were in the linear range. Densitometry measurements were performed with MCID analysis software 7.1 (InterFocus, Ltd) on sections for each region per mouse (4 mice per experimental group for a total of 16 mice).

VGLUT1, VGLUT3, VAChT and VGAT are vesicular transporters that mediate neurotransmission from the presynaptic side. VGLUT1 is necessary for the vesicular accumulation of glutamate and is a general marker for the glutamatergic drive in synapses. In the context of grid cells, the excitatory drive is a prerequisite for grid cell generation[28]. VGLUT3 is a specific marker for synapses made by CCK-positive basket cells in the MEC. VAChT expression in the MEC marks presynaptic cholinergic terminals from the medial septum. The severe loss of cholinergic neurons is a hallmark of Alzheimer's disease and a possible role of acetylcholine for grid cell activity has previously been reported. VGAT mediates vesicular accumulation of GABA and is a general marker for inhibitory drive which is important for grid cell generation[32,33]. In the case of NR1, it is a subunit of NMDA receptors that has previously been shown to be necessary for both the generation of grid cells and path integration ability[35].

**Reporting summary**. Further information on research design is available in the Nature Research Reporting Summary linked to this article.

## Data availability
All data supporting the key findings of this study are available within the article, Supplementary Information and Source Data, or via request to the corresponding author. Source data are provided with this paper.

## Code availability
All custom codes supporting the key findings of this study are available at the following GitHub page: https://github.com/johnson-ying/ying-et-al-2021, or via request to the corresponding author.

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

## Acknowledgements

We graciously thank S. Kim, Z. Ante, K. Harandian, Q. He, A. Ismailova, D. Patel, A. Zhen, and A. Milette-Gagnon for their assistance in experiments. We also thank J. Poirer, M. Hasselmo, J. Hinman, S. Villeneuve, S. Williams, R. Rozeske, J. Lee, J. Robinson, and E. Vachon-Presseau for comments on earlier versions of this manuscript and to all members of the Brandon laboratory for helpful discussions. This work was funded by CIHR Project Grants #367017 and #377074, an NSERC Discovery Grant #74105, a Scottish Rite Charitable Foundation Grant, a Canada Fund for Innovation Grant, and a Canada Research Chairs award to M.P.B. J.Y. is supported by a Doctoral Training Grant from the Fonds de recherche du Québec, and previously by a Master's Training Grant from the Fonds de recherche du Québec and a CIHR Master's Training Fellowship.

## Author contributions

J.Y. contributed to experimental design, recordings, analysis of data, and wrote the manuscript. R.L. contributed to immunohistochemistry quantifications. A.T.K. contributed to analysis of data. E.V. and S.E.M. contributed to immunoautoradiographic quantifications. M.P.B. contributed to experimental design, analysis of data, and wrote the manuscript.

## Competing interests

The authors declare no competing interests.
