## [Peer Review File · Nature Communications]

Disruption of the grid cell network in a mouse model of early Alzheimer's diseaseEditorial Note: This manuscript has been previously reviewed at another journal that is not operating a transparent peer review scheme. This document only contains reviewer comments and rebuttal letters for versions considered at *Nature Communications*.

REVIEWERS' COMMENTS

Reviewer #3 (Remarks to the Author):

This is the third time that I review this manuscript investigating spatial coding in the medial entorhinal cortex and the hippocampus in a model of the early stages of Alzheimer's disease. The main claims in the paper are well supported by the results and the data analysis is "state-of-the-art". The manuscript uncovers the main alterations in spatial codings present at an early stage of a mouse model of Alzheimer's disease. These alterations provide possible causes for the decline in navigational abilities associated with early AD.

The authors have addressed all my previous concerns adequately. I only have minor points that the authors might want to clarify in the manuscript.

1. Figure 3: The 2D displacement of the peak in the spatial cross-correlations is presented in C and D. Was the value at the peak within the box (region) also decreased for grid cells or is it just that the 2D shift of the peak value is larger?
2. Line 518: Was the 720x480 pixel resolution (4,9 pixels per cm) also used for the 100x100 cm? It seems that this would not cover the whole environment.
3. Line 565: Interneurons were identified only on their spike waveforms. If the database contains a sufficient number of fast-spiking interneurons, it would be interesting to know whether the reduced Grid-Interneuron coupling is also observed when considering only interneurons (narrow waveforms) with a firing rate larger than 5Hz.

Dear Reviewers,

Please accept this letter as a point-by-point response to the latest comments from our previous submission at *Nature Communications*. Each Reviewer comment is bolded, followed by our response. We greatly thank you for your continued support and guidance throughout this submission process.

Sincerely,

Mark Brandon, PhD
Assistant Professor
Canada Research Chair in the Neural Circuits of Memory
Department of Psychiatry
Faculty of Medicine
McGill University
Douglas Hospital Research Centre

**Reviewer #3:
Comments for the Author:**

This is the third time that I review this manuscript investigating spatial coding in the medial entorhinal cortex and the hippocampus in a model of the early stages of Alzheimer's disease. The main claims in the paper are well supported by the results and the data analysis is "state-of-the-art". The manuscript uncovers the main alterations in spatial codings present at an early stage of a mouse model of Alzheimer's disease. These alterations provide possible causes for the decline in navigational abilities associated with early AD.

The authors have addressed all my previous concerns adequately. I only have minor points that the authors might want to clarify in the manuscript.

1) Figure 3: The 2D displacement of the peak in the spatial cross-correlations is presented in C and D. Was the value at the peak within the box (region) also decreased for grid cells or is it just that the 2D shift of the peak value is larger?

It is just that the 2D shift of the peak value is larger, and we've added in a statement to emphasize this fact in the figure caption of Figure 3C: "Note that this analysis makes no conclusions about the magnitude of the peak correlation pixel, and strictly assesses the shift of said peak value."

2) Line 518: Was the 720x480 pixel resolution (4,9 pixels per cm) also used for the 100x100 cm? It seems that this would not cover the whole environment.

Yes, but the camera was placed at a height such that it covered the 100x100 cm box. We've added in the following statement in the methods section under "**Position, direction and velocity estimation**": "The camera was elevated at a height such that it fully captured all recording environment sizes used".

3) Line 565: Interneurons were identified only on their spike waveforms. If the database contains a sufficient number of fast-spiking interneurons, it would be interesting to know whether the reduced Grid-Interneuron coupling is also observed when considering only interneurons (narrow waveforms) with a firing rate larger than 5Hz.

Interneurons were, in fact, identified on both narrow waveforms, as well as a mean firing rate of at least 0.5 Hz (see "**Cell selection**" in the Methods section):

"Putative interneurons in the MEC were selected by having a narrow wave form (<0.3ms) and a mean firing rate of at least 0.5 Hz."

We re-ran our analysis on grid-interneuron pairs when only considering interneurons with a mean firing rate of 5Hz or larger. The results are shown in the bottom figure, along with the results presented in Main Figure 4 for comparison. This new selection criterion reduced our sample sizes to 3, 18, 11 and 9 pairs across nTG-y, nTG-a, APP-y and APP-a groups, respectively. Given a sample size of only 3 pairs, there is no point in running statistical comparisons across groups. For this reason, we opted not to present these graphs in the manuscript because they don't contribute any new knowledge, nor refute any of the claims that we made.

However, by visual inspection alone, the trend and pattern of these graphs are very similar to the data presented in the manuscript, particularly in the case of the APP-a group.